# Occurrence and Risk Factors Associated with *Platynosomum illiciens* Infection in Cats with Elevated Liver Enzymes

**DOI:** 10.3390/ani14071065

**Published:** 2024-03-30

**Authors:** Pinkarn Chantawong, Jiraporn Potiwong, Natchanon Choochote, Kakanang Piyarungsri, Chakorn Kunkaew, Sahatchai Tangtrongsup, Saruda Tiwananthagorn

**Affiliations:** 1Faculty of Veterinary Medicine, Chiang Mai University, Chiang Mai 50100, Thailand; pinkarn_@hotmail.com (P.C.); potiwong.pim@gmail.com (J.P.); nonchoochote@gmail.com (N.C.); kakananjp@gmail.com (K.P.); 2Research Center for Veterinary Biosciences and Veterinary Public Health, Faculty of Veterinary Medicine, Chiang Mai University, Chiang Mai 50100, Thailand; 3Research Center of Producing and Development of Products and Innovations for Animal Health and Production, Faculty of Veterinary Medicine, Chiang Mai University, Chiang Mai 50100, Thailand; 4Department of Animal Science and Fisheries, Faculty of Science and Agricultural Technology, Rajamangala University of Technology Lanna, Lampang 52000, Thailand; vetbird9@yahoo.com; 5Elephant, Wildlife, and Companion Animals Research Group, Faculty of Veterinary Medicine, Chiang Mai University, Chiang Mai 50100, Thailand

**Keywords:** *Platynosomum illiciens*, liver fluke, cats, liver enzymes, associated factors

## Abstract

**Simple Summary:**

Platynosomiasis is a feline hepatobiliary disease caused by *Platynosomum illiciens*, the most significant and widely distributed trematode in tropical and subtropical areas. In this study, we investigated the occurrence of *Platynosomum* spp. infection in cats with elevated serum alanine aminotransferase (ALT) levels and confirmed the species through DNA sequencing. Additionally, we assessed the association of factors and clinicopathological abnormalities with *Platynosomum* spp. infection to raise awareness and emphasize the importance of an appropriate deworming regimen to reduce the risk of *P. illiciens* infection.

**Abstract:**

*Platynosomum* spp., a hepatic trematode, causes fatal hepatobiliary disease in cats. Feline platynosomiasis is often underestimated due to a lack of awareness and diagnostic challenges. This study aimed to investigate the occurrence, factors, and clinicopathological abnormalities associated with *Platynosomum* spp. infection in cats with elevated serum ALT levels. *Platynosomum* infection was determined using zinc sulfate flotation and formalin–ether sedimentation. DNA sequence analysis of PCR products from the *Platynosomum* internal transcribed spacer 2 (ITS2) region and *cox1* gene was used to identify *Platynosomum* species. Of a total of 43 cat fecal samples, the proportion of *Platynosomum* spp. infection by microscopic examination was 11.63% (5/43). All PCR-positive samples were molecularly identified as *Platynosomum illiciens*. From the logistic regression analysis, the odds of *Platynosomum* infection in cats without a deworming program were 16 times higher than those of regularly dewormed cats. Demographic data, housing conditions, and predatory behavior were not significantly associated with the infection. Regarding blood profiles, infected cats had higher eosinophil counts (*p* = 0.014), with no significant differences in ALT (*p* = 0.791) or ALP (*p* = 0.970) levels compared to non-infected cats. Our findings demonstrate that eosinophilia in cats with increased serum ALT may suggest *P. illiciens* infection in endemic areas. We strongly recommend a regular deworming program to mitigate the risk of *P. illiciens* infection.

## 1. Introduction

Over the decades, helminthic infections in cats have gained the scientific interest of researchers due to the severity of their clinical manifestations [1]. The hepatic trematode *Platynosomum illiciens* (syn. *P. fastosum* and *P. concinnum*) is the most important and widely distributed parasite of cats in tropical and subtropical areas. The prevalence of this liver fluke has been reported to range from 0.07% to 81% worldwide [2]. In Asian countries, liver fluke infection in cats has been reported in Malaysia [3], Sri Lanka [4], Vietnam [5,6], Thailand [7,8,9], and Korea [10].

Feline platynosomiasis is considered a fatal disease; however, the information on this parasitic infection is scattered. Cats become infected by ingesting intermediate hosts (terrestrial isopods) or paratenic hosts (lizards, amphibians, and insects) containing metacercaria, which then migrate through the common bile duct, smaller biliary ducts, and the gall bladder, causing hepatobiliary problems such as cholangiohepatitis, cholangiocarcinoma, hepatic fibrosis, and bile duct obstruction, resulting in related clinicopathologic abnormalities [2,11]. Eosinophilia and an increased activity of the hepatic enzymes alanine aminotransferase (ALT) and alkaline phosphatase (ALP) have been detected in platynosomiasis cases. However, increased serum ALP concurrent with eosinophilia is sporadically reported [11,12,13]. Infected cats present a variety of clinical signs such as cachexia, vomiting, diarrhea, ascites, progressive jaundice, and death. Nonetheless, some cats remain asymptomatic. These signs vary according to the severity of the infection, the number of adult parasites, the duration of infection, and the immune response [14,15].

*Platynosomum* spp. infection can be diagnosed by coproparasitological evaluation, including zinc sulfate flotation and formalin–ether sedimentation techniques [16,17]. Additionally, molecular techniques such as polymerase chain reaction (PCR) assays and DNA sequence analyses of multiple target genes have been applied to identify *Platynosomum* species [18]. However, most cases of platynosomiasis have been diagnosed as incidental findings during necropsies due to its non-specific clinical effects [2,11]. This problem may lead to the underdiagnosis of this parasite and a lack of prompt treatment. To our knowledge, many reports have documented sudden deaths in platynosomiasis cases presenting only clinical pathology abnormalities, with a notable increase in hepatic enzymes [2,13,19]. Therefore, this study aimed to investigate the occurrence, risk factors, and clinical pathology abnormalities associated with *Platynosomum* infection in cats with elevated serum ALT levels. This fundamental study could increase awareness of this fatal disease among veterinary practitioners. 

## 2. Materials and Methods

### 2.1. Ethical Approval

The research protocols were approved by the animal ethics committee of the Faculty of Veterinary Medicine, Chiang Mai University (9 July 2021; Ref. No. S19/2564), under the guidelines for the Care and Use of Experimental Animals, National Research Council of Thailand.

### 2.2. Study Area and Animal Selection

The study area was Chiang Mai Province in the northern region of Thailand, at the coordinates 18°47′47.22″ N, 98°57′40.644″ E (Figure 1), and samples were collected at the Small Animal Veterinary Teaching Hospital, Faculty of Veterinary Medicine. Domestic cats with elevated serum ALT levels (over 97 U/L, reference range 25–97 U/L) [20] and without a history of acute traumatic conditions were selected from this hospital between January 2020 and October 2021. Inclusion criteria were as follows: cats aged over three months, regardless of breed, sex, pregnant or lactating status, and cat’s housing. No cats were treated with topical or systemic parasiticides in the 8 weeks preceding the study. The sample size of 43 cat fecal samples was calculated based on a 2.7% prevalence rate from a previous study conducted in an animal refuge in Nakhon Nayok, Thailand [8], with an acceptable relative error of 5% and a 95% confidence interval. The samples were randomly selected from cat populations to increase detection potential using a list of random numbers generated from the Open Epi program 3.01 [21].

### 2.3. Data and Sample Collection

The individual demographic data (age, sex, and breed) of 43 selected cats, as well as risk factors including the cats’ housing (indoor or outdoor), predatory behavior, and deworming regimens (time and frequency of deworming), were retrieved from the cats’ owners using a questionnaire. Clinical signs and blood profiles for each selected cat were recorded, including ALT, ALP, and eosinophil levels. Eosinophil count was performed using an automated hematological analyzer (BC-5300 Vet, Mindray, Shenzhen, China), and serum ALT and ALP assessments were performed using an automated chemistry analyzer (BX-3010, Sysmex Corporation, Tokyo, Japan) according to the manufacturer’s recommendations.

Forty-three fecal samples were collected from the litter boxes of individually housed cats. After that, fecal samples were transported on ice to the Parasitology Laboratory at the Faculty of Veterinary Medicine, Chiang Mai University, Chiang Mai, Thailand. The samples were stored at 4 °C and analyzed within 1 week.

### 2.4. Parasitological Procedures and Identification

#### 2.4.1. Fecal Analysis

All fecal samples were examined under a light microscope after simple flotation using zinc sulfate solution (specific gravity: 1.35) and formalin–ether sedimentation techniques [1,17]. A fecal sample was considered *Platynosomum*-positive if at least one *Platynosomum*-like egg, primarily identified using Basu et al.’s morphological keys [2], was identified. The remaining positive fecal samples were stored at −20 °C until further molecular analysis.

#### 2.4.2. Molecular Technique for Species Identification

Genomic DNA samples from *Platynosomum*-positive fecal samples were extracted using the NucleoSpin^®^ DNA stool kit (Macherey-Nagel GmbH, Düren, Germany) according to the manufacturer’s instructions.

Two PCR assays were conducted to identify the species of *Platynosomum* in the positive samples. DNA fragments of the internal transcribed spacer 2 (ITS2) region (~560 bp) and trematode mitochondrial cytochrome c oxidase 1 (*cox1*) gene (396 bp) were amplified using published primers and thermocycler programs [18,22]. A forward primer, ITS5 (5′-GAAGTAAAAGTCGTAACAAGG-3′), and a reverse primer, d58R (5′-CACGAGCCGAGTGATCCACCGC-3′), were used for the ITS2 amplification, and a forward primer, JB3 (5′-TTTTTTGGGCATCCTGAGGTTTAT-3′), and a reverse primer, Trem.*cox1*.rrnl (5′-AATCATGATGCAAAAGGTA-3′), for *cox1* amplification. An annealing temperature of 55 °C was used to amplify the ITS2 fragment [18], and 50 °C was used for the *cox1* fragment [22]. Distilled water (DW) and adult *Platynosomum illiciens* DNA harvested from an infected cat carcass were used as negative and positive controls, respectively. PCR-positive products were purified using a Nucleospin^®^ PCR Clean-up Kit (Macherey-Nagel GmbH, Düren, Germany) and submitted for direct sequencing using a commercially available direct sequencing service (Macrogen, Seoul, Republic of Korea) in the forward and reverse directions using the PCR primer sets described above. Only two *cox1* and four ITS2 nucleotide sequences of *Platynosomum* were successfully retrieved. Nucleotide sequences from forward and reverse directions were edited and manually aligned to retrieve a consensus sequence using BioEdit Sequence Alignment Editor Software version 7.7 [23]. Phylogenetic analysis was performed using MEGA X [24]. Available sequences of *Platynosomum* spp. from Vietnam [6], Brazil [18], and Sri Lanka [4] were used as reference sequences for tree construction. In addition, a sequence of *Dipylidium caninum* from Italy (MT806359) [25] and *Dicrocoelium dendriticum* from Japan (LC629058) [26] were used as an outgroup species for the *cox1* gene and ITS2 region, respectively. Multiple sequences were aligned using ClustalW, and phylogenetic analysis was performed using a maximum likelihood (ML) method based on the Tamura–Nei model [27]. A consensus tree was obtained after a 1000-replication bootstrap analysis.

### 2.5. Statistical Analysis

Demographic data, factors, clinical signs, and blood profiles were entered into a spreadsheet, and all statistical analyses were performed using Stata statistical software release 16.1 (StataCorp, College Station, TX, USA). The prevalence of *Platynosomum* infection in cats with elevated ALT levels was calculated with a 95% confidence interval (CI). Odds ratios (OR) and the 95% CI were estimated to measure the strength of the association using univariable logistic regression analysis. Variables associated with *Platynosomum* infection (*p* ≤ 0.1) were included in a multivariable logistic regression analysis. A multivariable logistic regression model was performed using a backward stepwise elimination procedure against the parasitic examination results. The normality of serums ALT and ALP and eosinophil levels was assessed using the Shapiro–Wilk test. Differences in ALT, ALP, and eosinophil levels between parasitized and non-parasitized cats were compared using the Mann–Whitney U test. *p*-values ≤ 0.05 were considered statistically significant.

## 3. Results

### 3.1. Occurrence of Platynosomum spp. Infection

A total of 43 fecal samples were obtained from cats of different ages, sexes, and breeds (Table 1). The study population comprised 65.12% female and 34.88% male cats; most of them were juvenile (1–7 years old, 72.1%) and domestic short-haired cats (74.42%). The proportion of *Platynosomum* spp. infection in cats with elevated ALT levels was 11.63% (95% CI: 4.70–25.58), and all were determined as microscopically positive in both the zinc sulfate flotation and formalin–ether sedimentation techniques. The single infection rate of this hepatic trematode was 4.65%, whereas dual infections with *Toxocara* spp. or *Ancylostoma* spp. occurred at 2.33%, and triple infections with *Toxocara* spp. and *Toxascaris* spp. were found at 2.33% (Table 2).

The *Platynosomum* eggs presented with a mean length of 32.34 ± 10.85 µm (range: 13.75–40.0 µm) and a width of 22.6 ± 7.27 µm (range: 10.0–30.0 µm). A varied size of typical eggs was predominantly observed in the fecal samples of four infected cats. Morphologically, they were golden to dark brown, elliptical, and embryonated with an operculum at one end (Figure 2A–D). In one infected cat, an atypical egg was smaller and lacked an embryo (Figure 2E).

### 3.2. Platynosomum Species Confirmation and Genetic Characteristics of P. illiciens Based on ITS2 Region and cox1 Gene

Four of the five *Platynosomum*-egg-positive samples were assessed using PCR assays targeting the ITS2 region and the *cox1* gene. One positive fecal sample was not available for PCR assays (Cat13-CMU). *Platynosomum* DNA was successfully amplified in all samples using ITS2 PCR assays, and in two samples using a *cox1* PCR assay. Nucleotide sequence analyses revealed that all *Platynosomum*-positive samples were *P. illiciens.* The sequences of *P. illiciens* in cat isolates were deposited in GenBank (DDBJ/EMBL/GenBank database Accession No. LC779845–LC779848 for ITS2 and LC779849–LC779850 for *cox1*). The *cox1* sequence in one cat (Cat1-CMU) exhibited 100% identity with *P. illiciens* found in a cat isolate from Brazil (GenBank: OM368257). The ITS2 sequence presented 99.74% identity with *P. illiciens* isolates from cats from Sri Lanka (GenBank: OK254044) and Costa Rica (GenBank: LC500530). As displayed in Figure 3A, the *cox1* sequence of one isolate, P1-COX1-Cat1-CMU, was placed in the same group of *P. illiciens* isolates from cats from Brazil (GenBank: MH155181-184), while P16-COX1-Cat16-CMU was separated in the different clade. Regarding the ITS2 sequences (Figure 3B), P9-ITS2-Cat9-CMU was close to *P. illiciens* isolates from cats from Brazil (GenBank: MH156565). Two isolates, P2-ITS2-Cat2-CMU and P16-ITS2-Cat16-CMU, were close to *P. illiciens* isolates from cats from Sri Lanka (GenBank: OK254044). One isolate, P1-ITS2-Cat1-CMU, was separated into a different clade.

### 3.3. Associations of Demographic Data, Clinical Signs, and Factors with Platynosomum Infection

The associations of demographic data, clinical signs, factors, and *Platynosomum* infection are summarized in Table 3. Univariable logistic regression analyses demonstrated no significant association of *Platynosomum* infection with age, sex, breed, cat’s housing, predatory behavior, or clinical signs, including anorexia, weakness, vomiting, and jaundice. On the other hand, the deworming program (*p* = 0.003) and deworming time (*p* = 0.003) were statistically significantly associated with the parasitic infection. For the multivariable logistic regression analysis, when controlling for jaundice, the odds of Platynosomum infection in cats without a deworming program were 16 times higher than those of regularly dewormed cats (Table 4). Interestingly, two out of five infected cats had a history of receiving deworming drugs but followed an irregular deworming program.

Regarding the systemic deworming drugs, praziquantel at 5 mg/kg combined with pyrantel embonate at 57.5 mg/kg was the most commonly used parasiticide (34.88%; 15/43), followed by pyrantel embonate at 5 mg/kg (9.3%; 4/43) and fenbendazole at 30 mg/kg (4.65%; 2/43). Spot-on parasiticides were sporadically used, including fipronil at 11.11–33.2 mg/kg combined with (S)-methoprene at 13.51–40 mg/kg and praziquantel at 11.11–33.2 mg/kg (9.3%; 4/43); and imidacloprid at 12.5–25 mg/kg combined with moxidectin 1.25–2.5 mg/kg (4.65%; 2/43). Additionally, nine of the forty-three cats (20.93%) were dewormed with unknown medications.

### 3.4. Comparisons of Blood Profiles between Cats with and without Platynosomum Infection

Comparisons of serums ALT and ALP and eosinophil levels in parasitized and non-parasitized cats are shown in Figure 4. In platynosomiasis cats, the median of ALT was 134 U/L (IQR: 101–773 U/L), ALP was 80 U/L (IQR: 50–80 U/L), and eosinophil was 1.45 × 10^3^ cells/µL (IQR: 1.00–3.27 × 10^3^ cells/µL); in non-infected cats, the median of ALT was 199 U/L (IQR: 115–277 U/L), ALP was 75.5 U/L (IQR: 52–101 U/L), and eosinophil was 0.57 × 10^3^ cells/µL (IQR: 0.37–1.04 × 10^3^ cells/µL)**.** Notably, the median eosinophil level in infected cats was significantly higher than in non-infected cats (*p* = 0.014). However, the two groups had no significant differences in levels of ALT (*p* = 0.791) and ALP (*p* = 0.970). 

## 4. Discussion

In this study, we estimated the occurrence of and assessed factors and clinicopathological abnormalities associated with *Platynosomum* infection in cats with elevated serum ALT levels that visited the Small Animal Veterinary Teaching Hospital, Chiang Mai University, Chiang Mai, Thailand. The infection of *Platynosomum illiciens* in cats with increased ALT levels was common at a proportion of 11.63%. However, detecting this liver fluke in the fecal samples of cats can vary depending on the population tested and diagnostic techniques. 

*Platynosomum* infection represents a significant concern in cat populations, with documented prevalence rates varying from 0.07% to 9.76% in Asia [3,4,5,6,7,8,9,10]. In Thailand, the prevalence of *Platynosomum* infection in stray cats has been reported at 0.07% (1/1485) in Bangkok, 2.67% (4/150) in Pathum Thani and Nakhon Pathom Provinces, and 2.7% (8/300) in Nakhon Nayok Province [7,8,9]. In contrast, the prevalence of this parasite in cats with elevated ALT levels in the current study was 11.63% (95% CI: 4.70–25.58), notably higher than previous reports in Thailand. Previous reports have revealed that *P. illiciens* infection is typically asymptomatic, with animals often harboring parasites without displaying specific clinical abnormalities [11,28]. In a recent systematic review by Silva et al., the predominant specific symptom of *Platynosomum-infected* cats was icterus (45.8%), whereas the general symptom was weakness (65.8%) [29]. In this study, 60% (3/5) of infected cats were asymptomatic, while 40% (2/5) exhibited jaundice and anorexia. Furthermore, there were no statistically significant associations between other clinical signs and the presence of *Platynosomum* infection (*p* > 0.05). These findings are consistent with several previous studies [11,28,30]. Therefore, the difference in the estimated prevalence of *Platynosomum* infection in this study compared to others in Thailand may be attributed to the fact that all cats in this study had elevated serum ALT levels, suggesting problems in the hepatobiliary system that pose a high risk for platynosomiasis. It is also important to note that the severity of clinical signs depends on several factors, including the fluke burden, the infection duration, and the animal’s immune system response [14,15].

Detecting *Platynosomum* infection is challenging. The detection of *Platynosomum* eggs can vary depending on the sources of the sample and diagnostic techniques. Using fecal detection, the prevalence has been reported to be as low as 0.07%, but the prevalence of this parasite in necropsies is reportedly 3.16 to 45% [1,2]. In addition, the study of Ramos et al. revealed that microscopic egg detection from bile samples was 1.6 times more sensitive than manual trematode collection during necropsy. Regarding diagnostic techniques, many gaps remain in the history of platynosomiasis. This indicates that the infection may have occurred sporadically or has been underestimated over time due to the inefficiency of diagnostic methods [1]. However, recent studies have revealed that distinct coproparasitological evaluations, including the FLOTAC technique, using a flotation solution with a specific gravity of 1.35, and formalin–ether sedimentation, are effective tools for diagnosing *P. illiciens* [1,2]. With detection under a light microscope after these concentrating techniques, false positives are unlikely; however, a false negative can result from a too-low specific gravity of flotation solution that cannot float *Platynosomum* eggs or from excessive debris in the sample that masks the detection of the egg [16]. In one study, the formalin–ether concentration technique was superior to the sugar flotation technique with a specific gravity of 1.27 [8]. In this study, fecal flotation using a flotation solution with a specific gravity of 1.35 showed a detection rate similar to formalin–ether sedimentation. Therefore, the flotation technique with a specific gravity solution of 1.35 or formalin–ether sedimentation may be suggested for detecting trematode eggs.

Upon examination of the *Platynosomum* eggs under light microscopy, we expected golden to dark brown, elliptical, and embryonated eggs containing an operculum at one end, with a size range of 20–35 × 34–50 μm [6,31,32]. The morphology of *Platynosomum* spp. eggs found in this study was primarily consistent with previous reports. However, *Platynosomum* eggs from one cat (Cat2) were smaller (10.0–13.75 × 13.75–20.0 μm) with a thick shell, long elliptical shape, and lack of internal germ balls. We speculate that the smaller size and unusual morphology of these eggs could point to the detection of immature, atypical, unfertilized, or abnormal eggs that have been previously reported [2,32,33]. As morphological examination cannot be used for species differentiation, in this study, the species of all Platynosomum-positive eggs were identified using nucleotide sequence analyses of products from PCR assays targeting the ITS-2 region and cox-1 gene. All positive samples were 100% identical to *Platynosomum illiciens.*

Regarding age, cats’ housing, and the predatory behaviors of cats, previous studies have reported a significant association with *P illiciens* infection, while sex, neutering status, and breed are not considered risk factors for platynosomiasis cases [14,15,29]. Cats older than two years have been shown to have a higher prevalence and higher odds of infection than cats aged less than 2 years [19]. Outdoor cats or free-roaming cats with hunting behaviors are at a higher risk of contracting parasitic infections, as they have increased accessibility to intermediate or paratenic hosts, which play an essential role in the life cycle of this parasite [31]. The present study demonstrated no significant associations between demographic data (age, sex, and breed), cats’ housing, or predatory behavior with *P. illiciens* infection. This finding may be due to the small sample size of cats in this study. However, we found that juvenile (1–7 years) outdoor cats with predatory behavior were more frequently infected, which aligned with findings from previous studies [14,19]. The higher prevalence rates in the juvenile group could be attributed to their carnivorous habits, which are typically expressed from one year of age. The prepatent period of this parasite is 56–60 days [27,34]. Although our findings did not demonstrate a statistically significant association with the infection, it is advisable to recommend keeping cats indoors as a preventive measure to reduce the risk of exposure to intermediate or paratenic hosts of *Platynosomum* spp.

The multivariable logistic regression analysis revealed an association between deworming programs and *P. illiciens* infection. Cats with no deworming schedule were 16 times more likely to be infected with *P. illiciens* than cats with a regular schedule. This result confirmed the findings of previous studies regarding the importance of a regular deworming program in decreasing the chances of infection [35,36]. Notably, two out of five infected cats underwent a deworming regimen but still were positive for *P. illiciens* infection. Of these two cats, one received a combination of pyrantel embonate at 57.5 mg/kg combined with praziquantel at 5 mg/kg, and another one received pyrantel embonate at 5 mg/kg for over a year without a repeat deworming program. Therefore, the infection may be due to these cats not receiving appropriate deworming, which could involve using an ineffective anthelmintic agent, underdosing, or irregular deworming. Regarding the treatment efficacy of anthelmintic medications for platynosomiasis, praziquantel at 20–30 mg/kg PO once daily for three to five consecutive days, with a repeat treatment 12 weeks after the initial medication, is the most commonly used method, reported to be more efficient than fenbendazole at 50 mg/kg PO twice a day for five days [36,37]. Therefore, we recommend regular deworming with praziquantel, accompanied by at least an annual fecal examination, to reduce the risk of infection, as well as monitoring for cats over one year of age that roam freely and lack a consistent deworming program. 

Regarding platynosomiasis in cats, signs of hepatocellular injury have been reported, including changes in serum ALT, with ALP usually remaining unaffected [11]. A marked increase in ALT indicates hepatic parenchymal cell damage, resulting in cytosol content leakage into the circulation. The elevation of ALP was associated with severe cholestasis, which was observed at a low frequency (16.6%) in platynosomiasis cases [28,38]. In this study, ALT and ALP showed no statistically significant differences between parasitized and non-parasitized cats. We speculate that these results were due to the small sample population of infected cats and the variability in the infectious status of the individual animals. The fluke burden and host immune status determined the severity of the parasitic infestation and the time the animals developed hepatic inflammation, leading to cholangitis or cholangiohepatitis related to clinicopathological abnormality consequences [11,28]. In this study, eosinophil counts were significantly higher in parasitized cats than in non-parasitized cats (*p* = 0.014). This result is entirely in agreement with other studies, in which eosinophilia has been reported to be more common in cats with platynosomiasis [39,40,41]. The study of Taylor et al. revealed that all cats experimentally infected with 125 flukes (small dose) and 1000 flukes (large dose) exhibited eosinophilia at peak level 4 to 5 months after infection [41]. It is well documented that helminths induce Th2-dominant immune responses and increase the number of eosinophils. The eosinophil chemotaxis has a direct cytotoxic effect on parasites [42].

Based on our findings, an elevation in eosinophil counts in cats with elevated serum ALT may be in accordance with *P. illiciens* infection. Nevertheless, a larger sample size is essential for a more comprehensive evaluation of the association between clinicopathologic abnormalities and the infection. Regarding other hepatobiliary enzymes, serum aspartate aminotransferase (AST) and gamma-glutamyl transferase (GGT) are interesting parameters for evaluating chronic hepatobiliary abnormalities. Therefore, further studies should investigate the changes in these enzyme parameters that might be associated with *P. illiciens* infection. Regarding the detection of fluke ova, using only coproparasitological evaluation was not robust enough to detect the infection. Fecal examination can only detect parasitic eggs during the shedding period. Thus, fecal samples should be collected and re-evaluated every 2–3 months due to the prepatent period of this fluke [28,34]. In platynosomiasis cases involving bile duct obstruction, the fluke ova cannot pass into the feces through the intestinal tract, potentially leading to misdiagnosis [11,31]. Therefore, additional diagnostic techniques, such as hepatobiliary ultrasonography and bile microscopic evaluation, should be investigated in further studies to enhance the diagnostic efficiency of *P. illiciens* infection in cats.

## 5. Conclusions

In conclusion, this study provided insights into the occurrence, factors, and clinicopathologic abnormalities associated with *P. illiciens* infection in cats with elevated liver enzymes. Our findings demonstrate that the presence of eosinophilia in cats, along with an increase in serum ALT levels, may suggest *P. illiciens* infection in endemic areas. These findings should raise awareness of this parasitic infection among veterinary practitioners. Furthermore, the implementation of an appropriate deworming program is strongly recommended as a preventive measure to minimize the risk of *P. illiciens* infection.

## Figures and Tables

**Figure 1 animals-14-01065-f001:**
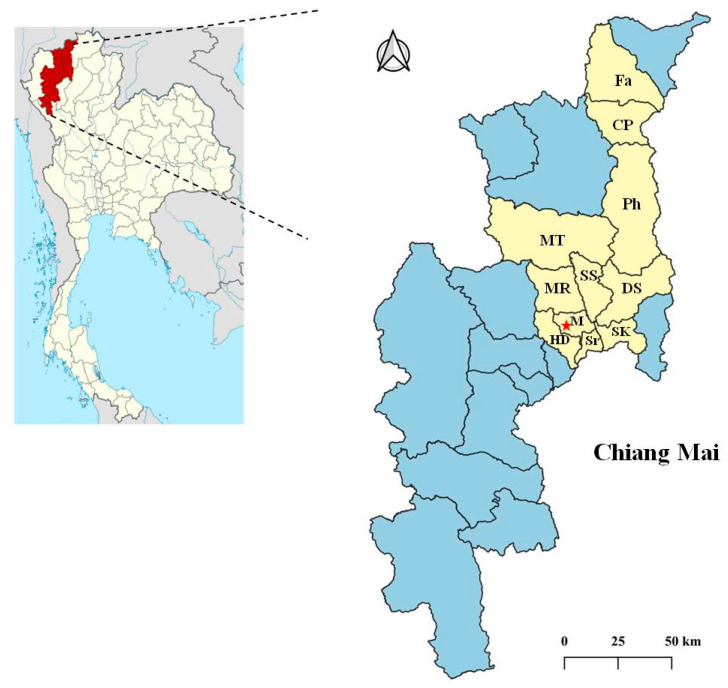
Map of the study area in Chiang Mai Province, Thailand. The star indicates the Small Animal Veterinary Teaching Hospital, Faculty of Veterinary Medicine. The 43 selected cats that reached the inclusion criteria resided in 11 districts, including Fang (Fa), Chai Prakan (CP), Phrao (Ph), Mae Tang (MT), Mae Rim (MR), San Sai (SS), Doi Saket (DS), Mueang Chiang Mai (M), Hang Dong (HD), Saraphi (Sr), and San Kamphaeng (SK).

**Figure 2 animals-14-01065-f002:**
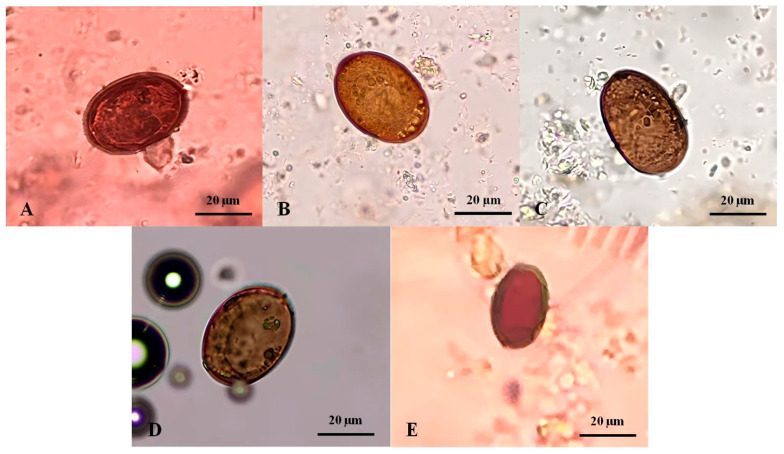
*Platynosomum* spp. eggs from infected cat fecal samples assessed by simple flotation method and examined under a light microscope. (**A**–**D**) *Platynosomum* spp. eggs were operculated, brown, and elliptical, containing undefined and granular embryos, and possibly germ balls inside. (**E**) Another form of *Platynosomum* spp. egg was smaller in size and brown, with a thick shell, long elliptical, and no internal germ balls.

**Figure 3 animals-14-01065-f003:**
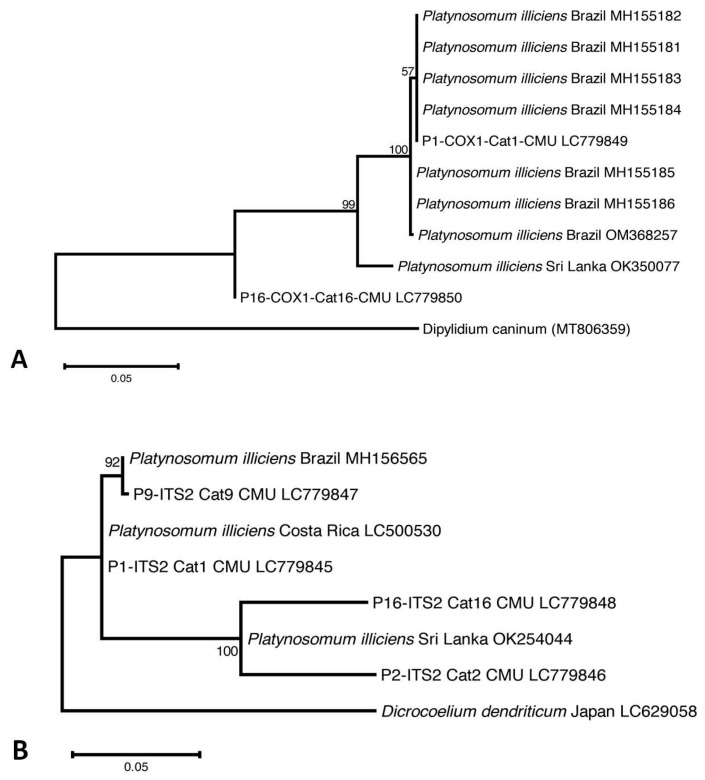
The evolutionary relationship of *P. illiciens* isolates from cats, based on 396-bp fragments of the mitochondrial cytochrome c oxidase subunit1 (*cox1*) gene (**A**), and on 396-bp fragments of the internal transcribed spacer 2 (ITS2) region (**B**). The tree was constructed using a maximum likelihood method based on the Tamura–Nei model using MEGA X software 10.2.6. The number in each branch indicates the percentage of 1000 bootstrap replications. Sequences obtained from GenBank are indicated by their accession numbers.

**Figure 4 animals-14-01065-f004:**
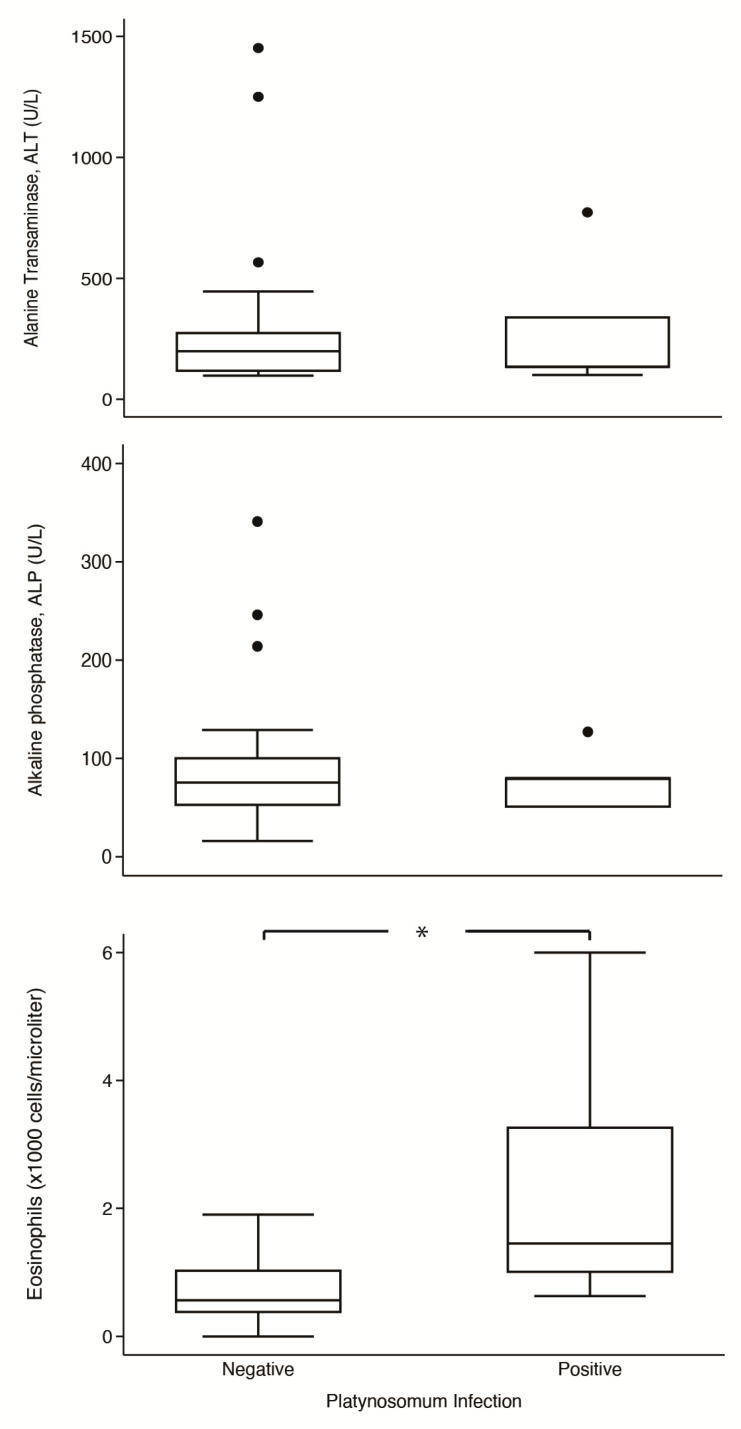
Box-plot comparison of alanine aminotransferase (ALT), alkaline phosphatase (ALP), and eosinophils counts between *Platinosomum*-positive and -negative cats. *, *p* < 0.05.

**Table 1 animals-14-01065-t001:** The demographic data (age, sex, and breed) of selected cat samples.

Demographic Data	Number (%)
Age	
Kitten (<1 year)	7 (16.28)
Juvenile (1–7 years)	31 (72.10)
Adult (7–14 years)	4 (9.30)
Geriatric (>14 years)	1 (2.33)
Sex	
Female	28 (65.12)
Neutered	19 (67.86)
Unneutered	9 (32.14)
Male	15 (34.88)
Neutered	6 (40.00)
Unneutered	9 (60.00)
Breed	
Domestic short-haired	32 (74.42)
Siamese	1 (2.32)
Persian	7 (16.28)
Scottish fold	2 (4.65)
British shorthair	1 (2.32)

**Table 2 animals-14-01065-t002:** Occurrence of *Platynosomum* spp. infection and co-infection of other helminths assessed by microscopic examination.

Helminthic Infections	No. Positive (% Positive)
Overall helminthic infections	14 (32.56)
*Platynosomum* spp.	5 (11.63)
*Ancylostoma* spp.	3 (6.98)
*Toxocara* spp.	8 (18.60)
*Toxascaris* spp.	2 (4.65)
*Spirometra* spp.	1 (2.33)
Single helminthic infection	10 (23.26)
*Platynosomum* spp.	2 (4.65)
*Ancylostoma* spp.	2 (4.65)
*Toxocara* spp.	5 (11.63)
*Spirometra* spp.	1 (2.33)
Dual helminthic infections	3 (6.98)
*Platynosomum* spp. + *Toxocara* spp.	1 (2.33)
*Platynosomum* spp. + *Ancylostoma* spp.	1 (2.33)
*Toxocara* spp. + *Toxascaris* spp.	1 (2.33)
Triple helminthic infections	1 (2.33)
*Platynosomum* spp. + *Toxocara* spp. + *Toxascaris* spp.	1 (2.33)

**Table 3 animals-14-01065-t003:** Univariable analysis of the associations between factors and *Platynosomums* infection in cats with increased alanine aminotransferase (ALT) (*n* = 43).

Variable	*Platynosomum*-Positive Number (%)	Odds Ratio	95% CI	*p* Value
Sex				0.643
Male	1/15 (6.70)	Reference		
Female	4/28 (14.3)	2.33	0.24–23.00	0.468
Neuter status				0.717
Female—intact	1/9 (11.10)	Reference		
Female—spayed	3/19 (15.80)	1.48	0.10–88.45	≅1.000
Male—intact	0/9 (0.00)	1.00	0–39.00	≅1.000
Male—castrated	1/6 (16.70)	1.55	0.02–141.06	≅1.000
Age				0.771
Kitten (<1 year)	0/7 (0.00)	Reference		
Juvenile (1–7 years)	5/31 (16.1)	1.67	0.20-inf.	0.677
Adult (7–14 years)	0/4 (0.00)	1.00	0.00-inf.	
Geriatric (>14 years)	0/1 (0.00)	1.00	0.00-inf.	
Breed				0.738
Persian	0/7 (0.00)	Reference		
Domestic short-haired	5/32 (15.60)	1.61	0.19-inf.	0.700
Siamese	0/1 (0.00)	1.00	0.00-inf.	
Scottish fold	0/1 (0.00)	1.00	0.00-inf.	
British shorthair	0/2 (0.00)	1.00	0.00-inf.	
Cat’s housing				0.589
Indoor	3/32 (9.40)	Reference		
Outdoor	2/11 (18.20)	2.11	0.15–21.62	0.760
Predatory behavior				0.145
Yes	5/28 (17.9)	4.09	0.15-inf.	0.204
No	0/15 (0.00)	Reference		
Deworming program				0.003
Regular	0/28 (0.00)	Reference		
Irregular	2/8 (25.00)	9.43	0.70-inf.	0.089
No deworming	3/7 (42.90)	20.68	2.00-inf.	0.011
Deworming time				0.003
2–6 months	0/24 (0.00)	Reference		
6–12 months	1/10 (10.00)	2.40	0.06-inf.	0.588
>12 months	1/2 (50.00)	12.00	0.31-inf.	0.154
No deworming	3/7 (42.90)	17.70	1.71-inf.	0.016
Clinical signs				≅1.000
No	3/25 (12.00)	Reference		
Yes	2/18 (11.11)	0.92	0.07–9.02	≅1.000
Anorexia				0.575
No	3/33 (9.09)	Reference		
Yes	2/10 (20.00)	2.44	0.18–25.41	0.657
Vomiting				≅1.000
No	5/37 (11.63)	Reference		
Yes	0/6 (0.00)	0.86	0–7.49	0.906
Jaundice				0.060
No	3/39 (7.69)	Reference		
Yes	2/4 (50.00)	10.72	0.59–202.65	0.120
Weakness				0.116
No	4/42 (9.52)	Reference		
Yes	1/1 (100.00)	7.60	0.19-Inf	0.233

**Table 4 animals-14-01065-t004:** Multivariable analysis of the associations between factors and Platynosomums infection in cats with increased alanine aminotransferase (ALT) (*n* = 43).

Factor	Odds Ratio	95%CI	*p* Value
Irregular deworming	5.10	0.30-Inf	0.259
No deworming	15.95	1.40-Inf	0.022
Jaundice	6.13	0.23–522.33	0.439

## Data Availability

The sequences of *P. illiciens* generated in the current research were submitted to GenBank (DDBJ/EMBL/GenBank database Accession No. LC779845–LC779848 for ITS2 and LC779849–LC779850 for *cox1*).

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
