# Peer review of "Occurrence and Risk Factors Associated with Platynosomum illiciens Infection in Cats with Elevated Liver Enzymes"

_animals, 2024, doi:10.3390/ani14071065_

Round 1
Reviewer 1 Report (Previous Reviewer 1)
Comments and Suggestions for Authors
The manuscript has been widely improved. It's well-written and understandable in the current form. Despite the low number of samples collected, results show a general perspective of Platynosomum sp. infection in cats with ALT increased
Material and methods:
· Data was collected by using a questionnaire. Who was filling the questionnaire? the owner or the vet?
· Please, re-writte: Line 84-86: The study area is the Small Animal Veterinary Teaching Hospital, Faculty of Veterinary Medicine, located in Chiang Mai province, the northern region of Thailand, at the 84 coordinates 18Ëš47’47.22”N, 98Ëš57’40.644”E (Figure 1).
The study area is the Chiang Mai province, the northern region of Thailand, at the coordinates 18Ëš47’47.22”N, 98Ëš57’40.644”E (Figure 1) and samples were collected at the Small Animal Veterinary Teaching Hospital, Faculty of Veterinary Medicine.
· line 93-94: Cats with elevated liver 93 enzymes are more likely to be infected with Platynosomum spp. [11, 13]. This statement should be in the Discussion section instead of Material and methods section.
Results
Any association among positive cats and deworming drugs? Were negative cats most routinely dewormed with Praziquantel ? Authors pointed out: Notably, 2 out of 5 347 infected cats underwent a deworming regimen but still were positive for P. illiciens in fection. (Line 347-348). In my opinion, it should be included in the Results section instead of Discussion section.
· Line 246-247. nine of the forty-three cats (20.93%) were dewormed with unknown medications. By whom was “unknown”? By the owner? Who filled the questionnaire? Was the owner who decided when and how dewormed his/her cat? Was considered regular deworming although it was unknown the drug used? Please, it should be explained and clarified.
Discussion
Line 265: we estimated the prevalence and assessed factors … I’d recommend the word occurrence rather than prevalence
Since most infections were related to eosinophilia rather than to liver damage. No statistical differences were observed when comparing ALT, and ALP levels. Authors speculated these results were due to the small sample population of infected cats but.. I’m wondering if most Platynosomum sp. infection are well tolerated by the cat? .
Would it be possible to quantify the parasite burden by coprology? McMaster?
Author Response
Please see the attachment.

Reviewer 2 Report (New Reviewer)
Comments and Suggestions for Authors
Dear Authors, thank you for submitting an interesting article entitled „Occurrence and factors associated with Platynosomum illiciens infection in cats with elevated liver enzyme” because the transfer of information is always useful.
The study investigated the occurrence, factors, and clinicopathological abnormalities associated with Platynosomum spp. infection in cats with elevated serum alanine aminotransferase (ALT) levels. Through a combination of microscopic examination and molecular analysis, the authors identified Platynosomum illiciens as the causative agent in infected cats. They found a significant association between the lack of a deworming program and Platynosomum infection, emphasizing the importance of regular deworming to mitigate the risk. Notably, infected cats exhibited higher eosinophil counts, suggesting eosinophilia as a potential indicator of Platynosomum infection in endemic areas.
Below you can find some general and specific comments.
General Comments:
The paper addresses an important aspect of feline health, shedding light on the underappreciated issue of Platynosomum spp. infection in cats. The study's focus on cats with elevated ALT levels adds clinical relevance, as hepatic trematode infections can often lead to hepatobiliary diseases. The methodology employed, including both microscopic examination and molecular analysis, strengthens the study's findings and contributes to our understanding of Platynosomum infection in cats. However, further clarification on the clinicopathological abnormalities associated with Platynosomum infection beyond eosinophilia would enhance the comprehensiveness of the study.
Specific Comments:
The methodology section provides a clear overview of the diagnostic techniques used for identifying Platynosomum infection.
While the study highlights the association between eosinophilia and Platynosomum infection, it is important to consider other potential clinicopathological abnormalities that may be indicative of hepatic trematode infestation. Further discussion on the implications of elevated ALT levels in infected cats and their correlation with disease progression would enrich the clinical relevance of the findings.
Line 2 - in the title I would insert risk before factors
Line 72 - please insert the term risk between investigate the occurrence and factors
Lines 83-85 – please rephrase the paragraph because the study area is not a veterinary hospital, it is the area from which the animals arrive at the hospital.
Line 90 – as you have specified inclusion criteria, it would also be necessary to specify exclusion criteria or a list of differential diagnoses of other pathologies with similar changes. Eosinophilia and elevated levels of liver enzymes such as alanine aminotransferase (ALT) or alkaline phosphatase (ALP) can be indicative of various diseases in cats. Some of the specific cat diseases associated with these findings include:
ü Feline Infectious Peritonitis can cause a wide range of symptoms. Eosinophilia and elevated liver enzymes can be seen in cats with FIP.
ü Cats with hepatic lipidosis may have elevated liver enzymes such as ALT and ALP, along with other signs such as anorexia and jaundice.
ü Cholangiohepatitis can be caused by infectious agents, immune-mediated processes, or other factors. Cats with cholangiohepatitis may have elevated liver enzymes and eosinophilia.
ü Toxoplasma gondii can affect the liver and other organs in cats and eosinophilia and liver enzyme abnormalities may be observed.
ü Pancreatitis can also affect the liver and cause elevated liver enzymes. In some cases, pancreatitis can lead to secondary hepatic lipidosis, further complicating the liver function.
ü Parasitic infections such as migrating larvae of certain worms, can cause liver inflammation and dysfunction, leading to elevated liver enzymes and possibly eosinophilia.
It's essential to note that while these conditions are associated with eosinophilia and elevated liver enzymes in cats, diagnostic confirmation typically requires additional testing.
Line 184 – Table 1 - Adding all the cats, from all age categories, results in a total of 42 cats, not 43 as stated above, but after. Please check your calculations.
Lines 188-189 - please indicate how many parasite eggs were measured and if you have an additional table with the records.
Lines 272-273 - repeating information, redundant phrase with lines 46-47.
The manuscript is generally well-written and structured, making it easy to follow.
Overall, the study provides valuable insights into the epidemiology and clinicopathological manifestations of Platynosomum spp. infection in cats. The combination of diagnostic methods employed and the focus on clinically relevant parameters contribute to the significance of the findings.
Author Response
Please see the attachment.

This manuscript is a resubmission of an earlier submission. The following is a list of the peer review reports and author responses from that submission.
Round 1
Reviewer 1 Report
Comments and Suggestions for Authors
This manuscript deals on parasite Platynosomum illiciens infection in cats. The main goals are focused on the prevalence, risk factors and clinical pathology abnormalities associated to Platynosomum illiciens. It's very well written but there are some gapos in the methodology.
The main gap of this manuscript is related to sampling. Authors calculated the sampling size on the basis on the parasite prevalence in a animal rescue shelter (2.7%). Understanding prevalence as the amount of disease in a known population…. Did this prevalence used for the sample size calculation refer to cat population in general or just to cats with elevated liver enzyme?
Just cats with elevated ALT were included in the study. But the sample size was too low. As a consequence, the number of positive samples were not quite large enough to be statistically analyzed. Thus, results do not allow to get the most relevant conclusions.
Regarding methodology, were all samples analysed twice? By sedimentation and by flotation technique? Did the authors get any concordance degree by both techniques? Any Kappa value was calculated when comparing results by both techniques?
Regarding deworming information in the questionnaire, it should be included which anthelminthic is used during the deworming of those cats …
Why did the authors compare ALT and ALP with parasite and not-parasite cats? In my opinion, it could be a bias, taking into consideration that just those cats with an ALT over 97U/L were enrolled.

Reviewer 2 Report
Comments and Suggestions for Authors
Materials and methods
Why did the authors use the formalin-ether technique? Please justify this. Here are the reference articles where this technique is used, but it is not good for detection.
(De Carli, 2007 G.A. De Carli (Ed.), Parasitologia clínica: seleção de métodos e técnicas de laboratório para o diagnóstico das parasitoses humanas [Clinical Parasitology: Selection of Methods and Laboratory Techniques in the Diagnosis of Human Parasites], Atheneu, São Paulo (2007), p. 906).
Rocha, N.O.; Portela, R.W.; Camargo, S.S.; Souza, W.R.; Carvalho, G.C.; Bahiense, T.C. Comparison of two coproparasitological 431 techniques for the detection of Platynosomum sp. infection in cats. Vet. Parasitol. 2014, 204(3-4), 392-395.
“As per reference no. 8, the sample size is not sufficient to determine prevalence epidemiologically”.
“Could you please provide the missing information regarding the source of the positive control?”
“It is important to include the map showing of sample collection in materials and methods section.”
“It is also important to add missing information about selection criteria.”
Line no. 97-98: “Clinical signs and blood profiles for each selected cat, including serum ALT, ALP, and eosinophil levels, were recorded and defined as clinicopathological aspects. “Which procedure is adopted to record these levels?”
Results:
“The image of PCR results is missing.”
“It is also important to include information about accession numbers. Results with the provided accession numbers are not found on NCBI, EMBL, and DDBL.”
“It is important to add the missing accession number of this study to the phylogenetic tree.”
“It is important to add the missing reference for the morphological identification key if available.”
Discussion:
Line No. 292-295: “As the morphological examination cannot be used for species differentiation, in this study, we have confirmed the species of all Platynosomum eggs through molecular analysis, and all were identified as Platynosomum illiciens. Please justify this statement.”
General comments
General comments are given in sticky notes of this study manuscript.
